# Tacrolimus Inhibits Human Tenon’s Fibroblast Migration, Proliferation, and Transdifferentiation

**DOI:** 10.3390/biomedicines13122956

**Published:** 2025-12-01

**Authors:** Woojune Hur, Jeongeun Park, Jae-Hyuck Lee, Ho-Seok Chung, Jin-A Shin, Hun Lee, Hungwon Tchah, Jae-Yong Kim

**Affiliations:** 1Department of Ophthalmology, Asan Medical Center, University of Ulsan College of Medicine, Seoul 05505, Republic of Korea; dnwnsgj@naver.com (W.H.); tmac1984@naver.com (J.-H.L.); chunghoseok@gmail.com (H.-S.C.); jina7316@naver.com (J.-A.S.); yhun777@gmail.com (H.L.); hwtchah@kimeye.com (H.T.); 2Department of Medical Science, University of Ulsan Graduate School, Seoul 05505, Republic of Korea; jung915eun@naver.com

**Keywords:** tacrolimus, human Tenon’s fibroblast, Smad-dependent pathway

## Abstract

**Background/Objectives**: We aimed to investigate the effects of tacrolimus on human Tenon’s fibroblast (HTF) migration, proliferation, and transdifferentiation in vitro. **Methods**: HTF cells were subcultured and serum-starved for 24 h before being treated with 10 ng/mL tacrolimus. After 1 h, 30 ng/mL platelet-derived growth factor (PDGF) or 10 ng/mL transforming growth factor beta-1 (TGF-β1) was administered to the HTFs. Migration, proliferation, and transdifferentiation were assessed using WST-1 assays, scratch-induced directional wounding, and western blot analysis. The involvement of the TGF-β signaling pathway was examined via western blotting to measure phosphorylated Smad2, Smad3, ERK, and Akt levels. **Results**: TGF-β1 and PDGF enhanced HTF migration, proliferation, and transdifferentiation, whereas tacrolimus inhibited these effects. Tacrolimus also inhibited the TGF-β1-induced upregulation of phosphorylated Smad2 and Smad3, suggesting its inhibitory effects occur through TGF-β1 signaling. **Conclusions**: Overall, tacrolimus can inhibit PDGF- and TGF-β1-induced HTF migration, proliferation, and transdifferentiation, primarily through the Smad-dependent TGF-β signaling pathway. To develop a new therapeutic modality, further longitudinal in vivo studies and human clinical trials are warranted.

## 1. Introduction

Severe conjunctival fibrosis, which is caused by an excessive transdifferentiation of fibroblasts to myofibroblasts during wound healing after glaucoma and recurrent pterygium surgery, is a cause of surgery failure [1,2,3]. Corneal opacity, which occurs in the corneas of some patients after ablation using an excimer laser, is also a side effect of the wound healing response [4,5,6]. Mitomycin C, which is commonly used in clinical practice to prevent these problems, effectively inhibits the fibrosis reaction around the wound; however, due to its strong antimetabolism action, it can cause various side effects and complications, such as scleritis [7], necrotizing keratitis [8], corneal edema [9], corneal endothelial cell loss [10,11], and fibrotic encapsulated bleb [12]. Therefore, it is important to find alternative drugs with fewer side effects than mitomycin C and with comparable effects to control the wound healing response. Efforts have been made to look for alternative drugs, and in 2009, a study reported that the subconjunctival injection of an anti-vascular endothelial growth factor, bevacizumab, inhibited scar formation and improved the survival rate of filtering blebs in a rabbit trabeculectomy model [13]. Moreover, it was reported in 2010 that bevacizumab had an antifibrotic effect on human Tenon’s fibroblast (HTF) cells in vitro [14].

HTF cells are key cells involved in subconjunctival wound healing. The wound healing reactions of HTF cells consist of proliferation, migration, and contraction in the wound area. Extracellular matrix (ECM) remodeling occurs through the synthesis of a new ECM. During wound healing, platelet-derived growth factor (PDGF) and transforming growth factor beta (TGF-β) are secreted. Fibroblast proliferation is activated by PDGF. At this time, the overgrown fibroblasts are transdifferentiated into myofibroblasts by TGF-β. These transdifferentiated myofibroblasts show a phenotype that increases the expression of alpha-smooth muscle actin (α-SMA) and the synthesis of collagen 1 and fibronectin in ECM proteins. In reality, on the side effects of fibrosis after excimer laser-assisted corneal ablation or trabeculectomy, the transdifferentiation of fibroblasts into myofibroblasts by TGF-β1 is a more important factor than fibroblast proliferation [15]. Moreover, inflammatory cytokines, such as IL-1 and tumor necrosis factor alpha (TNF-α), are secreted during wound healing [15,16,17].

Tacrolimus is a macrolide-based antibiotic commonly used as an immunosuppressive agent. Tacrolimus inhibits T cell proliferation and IL-2 production by inhibiting calcineurin [18,19]. Based on previous studies, tacrolimus inhibits the proliferation of cardiac fibroblasts [20] and human corneal fibroblasts [21]. Tacrolimus also has a significantly higher immunosuppressive effect than cyclosporine A, and it is mainly used as an oral or intravenous injection agent for the treatment of severe myasthenia gravis among ophthalmic diseases [22,23]. Moreover, tacrolimus ointment, which is primarily used for the treatment of atopic dermatitis, can be used as an eye drop to prevent its recurrence while reducing steroid use or as an alternative to treat refractory inflammatory ocular surface diseases, such as atopic and vernal keratoconjunctivitis and ocular graft-versus-host disease [24,25,26]. However, it has not been widely used clinically due to discomfort, such as a burning sensation and itching. Here, we reported that a tacrolimus-coated silicone plate is more effective in inhibiting inflammation and fibrosis compared with a mitomycin C-coated silicone plate and is associated with fewer adverse effects in a rabbit model [27]. To our knowledge, there have been no reports about the effects of tacrolimus on conjunctival fibrosis and wound healing responses associated with HTF cells. This study is unique in that it elucidates the antifibrotic effects and molecular mechanisms of tacrolimus, going beyond previous in vitro and animal model studies and directly using HTF derived from clinical patients. Because HTFs are a major cell type involved in subconjunctival wound healing and postsurgical scar formation, our HTF-based experimental results have relatively high clinical relevance [28]. Furthermore, beyond the simple confirmation of antiproliferative effects, this study suggests a mechanism by which tacrolimus comprehensively blocks cell migration, proliferation, and transdifferentiation by inhibiting Smad-dependent TGF-β signaling in a state activated by PDGF/TGF-β1 [29]. These mechanistic evidences suggest that topical tacrolimus administration strategies (e.g., tacrolimus-coated silicone plates or eye drops) may offer a promising alternative in terms of safety and efficacy compared to mitomycin C, which has a potent antifibrotic effect but has limited clinical application due to reported local complications (e.g., corneal endothelial cell damage and necrotic disease) [30]. The HTF-based results of this study will be useful for dose selection for future clinical applications and can provide direct evidence for the design of topical delivery platforms and safety assessments (e.g., surface irritation, pain, and epithelial regeneration effects) [30].

## 2. Materials and Methods

### 2.1. HTF Cell Culture

Human conjunctivae were isolated from 12 eyes donated by six expired or brain-dead individuals. The cause of death was trauma (three of six donors), cerebrovascular disease (two of six donors), and cardiovascular disease (one of six donors). All conjunctivae were harvested completely within 2 h after the death of the donor. The study protocol was approved by the institutional review board of the Asan Medical Center (IRB No. 2014-0428). The research followed the tenets of the Declaration of Helsinki, and informed consent was obtained from the appropriate family members.

The Tenon’s sac of the donor conjunctiva was put in a 1% penicillin/streptomycin (P/S) (15140122, Thermo Fisher Scientific, Waltham, MA, USA) Dulbecco’s Modified Eagle Medium: Nutrient Mixture F-12 (DMEM/F12) (A4192001, Invitrogen-Gibco Life Technologies, Inc., Karlsruhe, Germany) culture with 0.1% protease (T4455, Sigma-Aldrich, St. Louis, MO, USA), and this was reacted at 37 °C for 1 h. Then, the tenon sac was transferred to a 1% P/S DMEM/F12 culture medium and was scraped with a scraper (90020, SPL Life Sciences, Pocheon, Republic of Korea). Then, the tenon sac was discarded, and the culture medium was collected in a 15-mL tube and centrifuged at 1000 rpm for 5 min. The supernatant was discarded, and the subnatant was mixed with a 10% fetal bovine serum (A5256701, FBS; Invitrogen-Gibco-Life Science Technology, Inc.) DMEM + 1% P/S culture solution. Then, it was transferred to a new dish then incubated at 37 °C and 5% CO_2_. After 24 h, it was transferred to a new 10% FBS DMEM + 1% P/S culture medium and sub-cultured once every 2–3 days as the cells grew. The number of passages increased as much as the number of subcultures; however, only cells within passage number 10 were used in the experiment. In this way, human tenon sac fibroblasts were obtained from the donor tenon sacs.

### 2.2. WST-1 Cell Viability and Proliferation Assay System

Water-soluble tetrazolium salt (WST-1) (11644807001, Roche Applied Science, Penzberg, Germany) is a product used to quantify cell proliferation capacities or cell viability. A chromogen named formazan generated from WST-1 was measured through an enzyme-linked immunosorbent assay (ELISA) microplate reader. It is caused by succiatete tetrazolium reductase, a dehydrogenase present in the mitochondrial electron transport system of metabolically active cells. It is also effective only in living cells. As the color intensity is linearly correlated with the number of cells, cell proliferation can be easily measured.

HTF cells were cultured in a 96-well plate, which is primarily used when performing the WST-1 assay, and serum starvation was performed after 24 h. Then, tacrolimus was injected. After 1 h, PDGF-BB or TGF-β1 was applied. WST-1 assay was performed after 72 h (day 6), and after 2 h, cell viability was evaluated by measuring the absorbance at 450 nm using an ELISA microplate reader (Model 680, Bio-Rad Laboratories, Hercules, CA, USA) with a reference wavelength of 600–650 nm.

### 2.3. Experiment Procedures

Human tenon sac fibroblasts were injected into 96-well (WST-1 analysis) or six-well plates (western blot analysis) and grown with 10% FBS DMEM + 1% P/S culture. After 24 h, the cultures were replaced with 1% FBS DMEM + 1% P/S culture and treated with serum starvation. Because the serum contains many growth factors and essential proteins, serum starvation was performed so that the substances in the serum do not react specifically with or inhibit the substances to be tested. Cell migration, proliferation, and transdifferentiation and TGF-signaling experiments were performed by applying 10 ng/mL tacrolimus (F4679; purity ≥ 98% [HPLC]; Sigma Aldrich, Inc., St. Louis, MO, USA) for 24 h after the serum starvation and injecting 30 ng/mL PDGF or 10 ng/mL TGF-β1 after 1 h. Each experiment was repeated three or more times, and the average value was used.

### 2.4. Scratch-Induced Directional Wounding (Migration) Assay

On the first day of the experiment, HTF cells were seeded in a six-well plate. On the second day, serum starvation was performed for 24 h in a 1% FBS DMEM culture medium. On the third day, wounding was performed by scraping the HTF cells attached to the bottom of the plate in a straight line using a microtip (2450-1K0, Eppendorf AG, Hamburg, Germany). The cells that were scraped off and had fallen off the floor could stick back to the floor over time; therefore, washout was performed thrice with fresh culture media, and the culture was refilled. Tacrolimus was treated, and PDGF-BB and TGF-β1 were applied an hour later. Cell migration was examined at 24 h after the tacrolimus treatment.

### 2.5. Western Blot for Cell Proliferation and Transdifferentiation

On the first day of the experiment, HTF cells were seeded in a six-well plate. On the second day, serum starvation was performed for 24 h in a 1% FBS DMEM culture medium. On the third day, tacrolimus (FK-506, ALX-380-008; Enzo Life Sciences, Inc., Farmingdale, NY, USA) was treated, and PDGF-BB (P3201, Sigma-Aldrich, St. Louis, MO, USA) and TGF-β1 (ENZ-PRT221; Enzo Life Sciences, Inc.) were applied an hour later. A protein sample was obtained at 24 (day 4) and 48 (day 5) h after the tacrolimus treatment, and western blotting was performed. Then, α-SMA, a cell differentiation marker, and proliferating cell nuclear (PCNA), a proliferation marker, were used as antibodies. PCNA did not respond to PDGF-BB, and TGF-β1 was used as a control.

### 2.6. Western Blot Assay

HTF cells were injected into a six-well plate, and we changed them to 1% FBS DMEM + 1% P/S culture after 24 h. After 24 h, 10 ng/mL tacrolimus (FK-506, ALX-380-008; Enzo Life Sciences, Inc., Farmingdale, NY, USA) was added. After 1 h, PDGF was applied. After 24 h, 10 ng/mL TGF-β1 (ENZ-PRT221; Enzo Life Sciences, Inc.) was injected, and 48 h later, the culture was removed and washed thrice using a PBS solution (10010031, Gibco, Inc., Waltham, MA, USA). HTF cells were then lysed by the addition of a lysis solution (ab152163, Abcam, Cambridge, UK) and 25 mM NaF (201154, Sigma-Aldrich, St. Louis, MO, USA) & 2 mM Na_3_VO_4_ (450243, Sigma-Aldrich, St. Louis, MO, USA) as phosphatase inhibitors, and, 1 mM PMSF (P7626, Sigma-Aldrich, St. Louis, MO, USA) as protease inhibitors. All cells were collected, which were then transferred to a 1.5-mL tube and centrifuged at 4 °C and 13,000 rpm for 15 min. After the centrifugation, the supernatant was transferred to a new tube, and the protein amount was quantified by Bradford protein-detection assay. The quantified protein was separated by SDS-polyacrylamide gel electrophoresis and was transferred to a nitrocellulose membrane. Monoclonal mouse antibodies against α-SMA (1:1000; A2228, Sigma-Aldrich, Inc.), PCNA (1:1000; 2586, Cell Signaling Technology, Inc., Beverly, MA, USA), Smad2 (1:1000; 3103, Cell Signaling Technology, Inc.), phosphorylated Smad2 (pSmad2)(1:1000; 3104, Cell Signaling Technology, Inc.), Smad3 (1:1000; sc-101154, Santa Cruz Biotechnology, Inc., Dallas, TX, USA), phosphorylated Smad3 (pSmad3)(1:1000; sc-517575, Santa Cruz Biotechnology, Inc.), ERK (1:1000; sc-271269, Santa Cruz Biotechnology, Inc.), phosphorylated ERK (pERK)(1:1000; sc-81492, Santa Cruz Biotechnology, Inc.), Akt (1:1000; sc-5298, Santa Cruz Biotechnology, Inc.), phosphorylated Akt (pAkt)( 1:1000; sc-377556, Santa Cruz Biotechnology, Inc.), and glyceraldehyde 3-phosphate dehydrogenase (GAPDH)(1:1000; 5174, Cell Signaling Technology, Inc.) were added to the previously transferred membrane and allowed to stand at 4 °C overnight. The next day, after washing the membrane with blocking solution, horse radish peroxidase-conjugated secondary antibodies (1:5000; 31430, Invitrogen-Gibco-Life Science Technology, Inc.) were added and reacted at room temperature for 1 h. After 1 h, after washing the membrane thrice with a blocking solution, it was transferred to an enhanced chemiluminescence detection system (sc-2048, Santa Cruz Biotechnology, Inc.). The band area and intensity of the exposed film were analyzed using densitometric scanning, and the mean intensity was quantified using an Image J software (version 2.1.0/1.53c; http://imagej.nih.gov/ij/ (accessed on 15 January 2025); provided in the public domain by the National Institutes of Health, Bethesda, MD, USA). Normalization was done by calculating the mean intensity ratio of the target protein to that of GAPDH.

### 2.7. Relevance to the TGF-β Pathway

On the first day of the experiment, HTF cells were seeded in a six-well plate. On the second day, serum starvation was performed for 24 h in a 1% FBS DMEM culture medium. On the third day, tacrolimus was treated, and PDGF-BB and TGF-β1 were applied after 1 h. Protein samples were obtained at 18 and 24 h after the tacrolimus treatment; then, western blotting was performed. The TGF-β signal transduction pathway was investigated via western blot analysis in relation to transdifferentiation, which was considered to be the most important in wound fibrosis. pSmad2, Smad3, ERK, and Akt were individually normalized with their total amounts.

### 2.8. Statistics Analysis

All quantitative experiments were performed independently at least three times, and results are presented as the mean ± standard deviation (mean ± SD). Statistical analysis was performed using SPSS v16.0 (SPSS Inc., Chicago, IL, USA) using a two-tailed test. Normality (Shapiro-Wilk) and homoscedasticity (Levene) were checked for intergroup comparisons. If normality was not met or the sample size was small, the nonparametric Kruskal–Wallis H test was used, and if significant, a Bonferroni correction was performed post hoc. Repeated measures over time were analyzed using the Friedman test, specifically to determine differences between Week 1 and Week 4 experimental results. If significant, a Wilcoxon signed-rank test with Bonferroni correction was used for post hoc analysis. Correlation between continuous variables was performed using Pearson correlations when normality was met, and Spearman correlations when non-normality was present. All multiple comparisons were subject to Bonferroni correction, with a post-adjustment *p* < 0.05 threshold for statistical significance. Raw and adjusted *p*-values are presented in the report.

## 3. Results

### 3.1. WST-1 Assay: Cell Viability and Proliferation

The HTF cells applied with PDGF-BB or TGF-β1 had significantly higher cell viability than those without the growth factors (*p* < 0.05; Figure 1A,B). Although the HTF proliferation induced by PDGF-BB or TGF-β1 was inhibited by all tacrolimus concentrations in a dose-dependent manner, there was a significant difference in inhibitory effect at concentrations ≥ 100 ng/mL (*p* < 0.05; Figure 1A,B).

### 3.2. Cell Migration

HTF migration was significantly increased by PDGF-BB and TGF-β1 compared with the untreated group (*p* < 0.05; Figure 2A,B). In the group treated with 10 ng/mL tacrolimus, HTF migration decreased, indicating that the HTF migration induced by PDGF-BB and TGF-β1 was significantly inhibited by tacrolimus (*p* < 0.05; Figure 2A,B).

### 3.3. Cell Proliferation and Transdifferentiation

In the western blot assay to investigate cell proliferation and transdifferentiation, only 10 ng/mL tacrolimus was used. The 30 ng/mL PDGF-treated group showed a significantly higher PCNA expression at 24 and 48 h than the untreated group (*p* < 0.05; Figure 3A,C), suggesting that the HTF proliferation was induced by PDGF. In the group treated with 10 ng/mL tacrolimus, PCNA expression was significantly decreased, indicating that the HTF proliferation increased by PDGF was inhibited by tacrolimus (*p* < 0.05; Figure 3A,C).

The expression of α-SMA was significantly higher at 24 and 48 h in the TGF-β1-treated group than in the TGF-β1-untreated group (*p* < 0.05; Figure 3B,D), so the transdifferentiation of HTF cells was induced by TGF-β1. In the group treated with 10 ng/mL tacrolimus, α-SMA expression was decreased, indicating that the transdifferentiation of HTF cells induced by TGF-β1 was inhibited by tacrolimus.

### 3.4. Relevance to TGF-β Signaling Transduction Pathway

The western blot analysis revealed that the levels of pSmad2, pSmad3, pERK, and pAkt were significantly increased in HTF cells treated with TGF-β1 (all *p* < 0.05; Figure 4A–C). Tacrolimus treatment inhibited the pSmad2 and pSmad3 elevation (all *p* < 0.05; Figure 4A,B), but it did not inhibit the pERK and pAkt increase at 18 and 24 h (Figure 4A,C). In addition, Figure 5 summarizes changes in p-Smad2/3 activation and subsequent cellular responses in the untreated and tacrolimus-treated groups after TGF-β1 and PDGF stimulation. This graphically illustrates that tacrolimus inhibits the phospho-Smad2/3 directly, thereby reducing cell proliferation, migration, and transdifferentiation.

## 4. Discussion

Presently, tacrolimus is widely used in ophthalmology, but its indications are usually different from this study. Tacrolimus is effective in allergic eye diseases [31,32,33,34], Thygeson’s superficial punctate keratitis [35,36], corneal endothelial rejection [37], blepharitis [38], or even birdshot retinochoroiditis [39]. However, in most cases, it is used as a steroid-sparing agent and not as an alternative to mitomycin C, as in this study. This could be the novelty and strength of this study. Our findings demonstrate that tacrolimus primarily modulates the TGF-β-induced Smad signaling pathway, effectively suppressing fibrotic responses by downregulating TGF-β1/SMAD2/3 expression. This positions it as a potent therapeutic candidate for corneal and conjunctival injuries. Tacrolimus’s Smad pathway modulation, consistent with its reduction of pro-fibrotic gene expression and EMT inhibition, suggests an innovative antifibrotic strategy [27]. If its efficacy with fewer mitomycin C-like side effects was confirmed and integrated with sustained-release coating technology, tacrolimus could prevent post-ocular surgery fibrotic complications, such as a coating for intraocular drainage implants [40]. Tacrolimus holds a unique advantage among calcineurin inhibitors and antifibrotic agents; it shows more potent effects than cyclosporin A and, crucially, offers a safer alternative to cytotoxic mitomycin C [41]. By selectively modulating the Smad pathway to suppress fibrosis without significantly affecting cell viability, tacrolimus presents a new paradigm for ophthalmologic postoperative fibrosis strategies. If it is confirmed that tacrolimus has fewer side effects than mitomycin C while effectively inhibiting the fibrosis reaction around the corneal and conjunctival wound, and if a tacrolimus slow-releasing drug coating technology is developed [42], it may also be useful. For example, sirolimus is used to prevent restenosis due to fibrosis after a coronary stent procedure. Tacrolimus may also be used as a coating drug for drainage implants, which are primarily used to prevent stenosis due to fibrosis in patients with glaucoma unresponsive to antiglaucoma eyedrops. To investigate the in vitro effects of tacrolimus on the migration, proliferation, and transdifferentiation of HTFs, HTF cells were serum-starved for 24 h. To do this, we first determined the non-toxic range through a preliminary WST-1 cell viability assay (measured for 24 h) and then determined the effective concentration that induced a clear biological response (proliferation, migration, and transdifferentiation). Furthermore, serum starvation, a standard approach to synchronize cell cycle and basal signaling, was used to enhance the sensitivity and clarity of the observed growth factor-induced effects.

Thus, cell survival was investigated using the WST-1 method, where WST-1 was measured 24 h after drug treatments, and drug cytotoxicity could be determined. Moreover, with the outcomes of various drug concentrations, it was feasible to find a concentration with cytotoxicity, and a concentration without toxicity was found and used in the experiment. Cell proliferation could be determined by measuring WST-1 after 48–72 h. In this study, the cell proliferation induced by PDGF and TGF-β1 and inhibited by tacrolimus was investigated (Figure 1). Although the WST-1 assay is an experimental method that can reveal both cytotoxicity and proliferation, it is nonspecific.

In the western blot analysis, which analyzed the expression level of the cell proliferation marker PCNA, tacrolimus was found to inhibit the cell proliferation induced by PDGF (Figure 3). Using rabbit Tenon’s capsule fibroblasts (TF), Wang et al. showed that tacrolimus reduced the proliferation and promoted the apoptosis of TFs by inhibiting the expression of survivin [43].

Using human TF from pterygium, Carvalho et al. revealed that tacrolimus exposure led to a significantly lower TF proliferation rate after one day of exposure in vitro [44]. Although these findings regarding the inhibitory effect of tacrolimus on fibroblast proliferation were consistent with previous studies, HTFs were notably used in this study. However, it showed that cell proliferation resumed over time, and by day 19, the group exposed to tacrolimus had a greater cell proliferation than the group untreated with tacrolimus [44]. Therefore, additional measurements after a period longer than 48 h are needed to determine the optimal exposure frequency of tacrolimus.

In the wound healing assay (migration assay), it was confirmed that the cell migration increased by the growth factors was inhibited to some extent by tacrolimus (Figure 2). However, the overall cell migration was not prominent, including that in the control group, which was considered to be less mobile because all experiments were conducted after the serum starvation. If the cell migration experiment was additionally performed with 10% serum rather than serum starvation, it might be comparable with the outcomes of the serum starvation.

It was mentioned that the transdifferentiation to myofibroblasts played a more important role than fibroblast proliferation in the side effects of fibrosis. Therefore, note that tacrolimus showed an inhibitory effect on the amount of α-SMA, a differentiation marker increased by the growth factors (Figure 3).

The TGF-β signaling pathway is involved in many cellular processes, including cellular growth, death, homeostasis, differentiation, and other functions in both adult organisms and embryos [45,46]. The cellular process regulated by the TGF-β signaling pathway is very extensive, but the process is relatively simple. TGF-β induces the Smad signaling pathway which controls target gene transcription [45]. A TGF-β superfamily ligand binds to a type II receptor to use and phosphorylate a type I receptor [47]. Then, the type I receptor phosphorylates a receptor-regulating R-Smad (Smad2/3) and Smad2/3 binds to a co-Smad, Smad4. When the Smad2/3 and Smad4 complexes accumulate in the nucleus, they act as transcription factors and participate in the regulation of target gene expression. This process is called the Smad-dependent pathway, including pSmad2/3 and pSmad4. In contrast, in the TGF-β signaling pathway, there are other Smad-independent pathways, such as pERK1/2, pAkt, and pJNK1/2. These pathways cooperate with Smad-mediated gene expression and have non-transcriptional roles such as cytoskeletal reorganization and motility, dissolution of epithelial junctions, and translational control [45]. In this in vitro experiment, tacrolimus inhibited the elevation of pSmad2 and pSmad3 increased by TGF-β1; however, there were no significant differences in pERK (extracellular signal-regulated kinases) and pAkt proteins (Figure 4A,C) between the tacrolimus-treated group and the TGF-β1-treated group (Figure 5). Thus, the prohibitory effect of tacrolimus on cell migration, proliferation, and differentiation occurred through the Smad2 and Smad3 pathways, and the relationship with Smad-independent signals (pERK and pAkt) was considered to be not significant. It shows that tacrolimus primarily inhibits phosphorylation of Smad2/3 by type II receptor rather than non-Smad pathways.

In this study, we demonstrated that the inhibitory effects of tacrolimus on HTF migration, proliferation, and transdifferentiation through various methods. However, as we mentioned, to gain more pronounced cell migration, it was preferable to perform an additional experiment with 10% serum.

Moreover, our findings showed that while tacrolimus inhibits conjunctival HTF fibrosis, it may simultaneously inhibit proliferation and migration, potentially leading to delayed wound healing. Tacrolimus can be used primarily for its anti-fibrotic effects in situations such as glaucoma valve implantation and recurrent pterygium excision. However, supportive additives should be considered to promote wound healing.

There are several limitations in this study. First, the mechanism by which tacrolimus inhibited HTF migration was not fully investigated in this study. Second, there is a paucity of positive control groups in which, for example, mitomycin C is used. Third, there is a lack of data concerning the side effects in long-term treatment, optimal concentration, and duration of treatment [27,31]. Thus, further investigations are needed to obtain more information before human clinical trials due to the severe drug toxicity of tacrolimus. It is necessary to conduct a study over a longer period. Furthermore, future trials are required to determine an appropriate method of drug delivery to reduce side effects such as discomfort or systemic effects.

Although tacrolimus holds great potential for ophthalmic applications, local side effects (e.g., stinging, burning sensation) and the potential for systemic absorption require clinical consideration [48]. Concerns about serious side effects, such as systemic immunosuppression and nephrotoxicity, are raised, particularly with long-term administration. To mitigate these side effects and maximize therapeutic efficacy, the following strategic approaches are essential. First, the development of improved formulations and drug delivery systems is essential [49]. Extended-release eye drops, micro/nanoparticles, or biocompatible implants can help maintain drug retention in the eye, reducing local irritation and systemic absorption and enhancing ease of administration. Second, studies optimizing the concentration and dosing frequency should be conducted to find a balance that maintains the antifibrotic effects of tacrolimus while minimizing side effects [50]. Given the wide range of concentrations and dosing frequencies being studied, this will form the basis for developing personalized treatment protocols for each patient. Finally, patient monitoring and education are crucial. Patients should be fully informed of potential side effects and instructed to immediately consult a healthcare professional if they experience any adverse effects [51]. When administering long-term, safety must be ensured through regular monitoring, including blood tests, to ensure that no systemic side effects are induced.

## 5. Conclusions

Tacrolimus effectively inhibited the migration, proliferation, and transdifferentiation of human Tenon’s fibroblasts induced by TGF-β1 and PDGF in vitro. These effects appear to be mediated through inhibition of the TGF-β1 signaling pathway, as evidenced by decreased phosphorylation of Smad2 and Smad3. These findings suggest that tacrolimus may have therapeutic potential in modulating fibroblast activity in fibrotic ocular conditions. These results suggest that sustained-release formulations or drug-expressing implant coatings using tacrolimus may be practical alternatives for ophthalmic scar suppression and neovascularization treatment. Long-term pharmacokinetic and toxicity evaluations in animal models and clinical trial design are needed in the future.

## Figures and Tables

**Figure 1 biomedicines-13-02956-f001:**
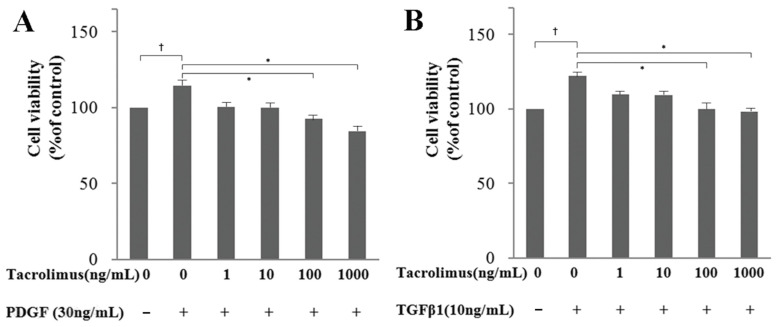
WST-1 assay to evaluate HTF cell viability and proliferation after tacrolimus treatment. (**A**) Effect of tacrolimus on PDGF-BB-induced HTF proliferation. PDGF-BB significantly stimulated HTF proliferation (*p* < 0.05). Tacrolimus dose-dependently inhibited this proliferation, with significant effects at concentrations ≥ 100 ng/mL (*p* < 0.05; n = 4). (**B**) Effect of tacrolimus on TGF-β1-induced HTF proliferation. Similarly, TGF-β1 significantly increased HTF proliferation (*p* < 0.05), which was also dose-dependently inhibited by tacrolimus (significant at concentrations ≥ 100 ng/mL, *p* < 0.05; n = 4). *: Significant difference between the proliferated group and the tacrolimus-treated proliferated group. †: Significant difference between the untreated group and the growth factor (PDGF-BB or TGF-β1)-treated group.

**Figure 2 biomedicines-13-02956-f002:**
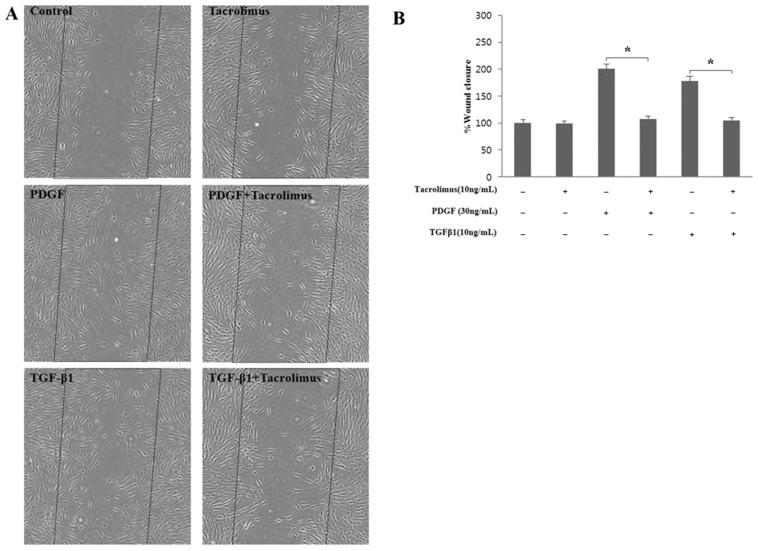
Scratch-induced directional wounding (migration) assay. Dashed lines indicate the wound area (**A**). HTF migration was significantly increased by PDGF-BB and TGF-β1 compared with that in the untreated group (**B**); *p* < 0.05; n = 4). In the group treated with 10 ng/mL tacrolimus, HTF migration decreased, indicating that the HTF migration induced by PDGF-BB and TGF-β1 was significantly inhibited by tacrolimus (*p* < 0.05; n = 4). *: significant difference between the induced group and tacrolimus-treated induced group. Scale bar = 10 mm.

**Figure 3 biomedicines-13-02956-f003:**
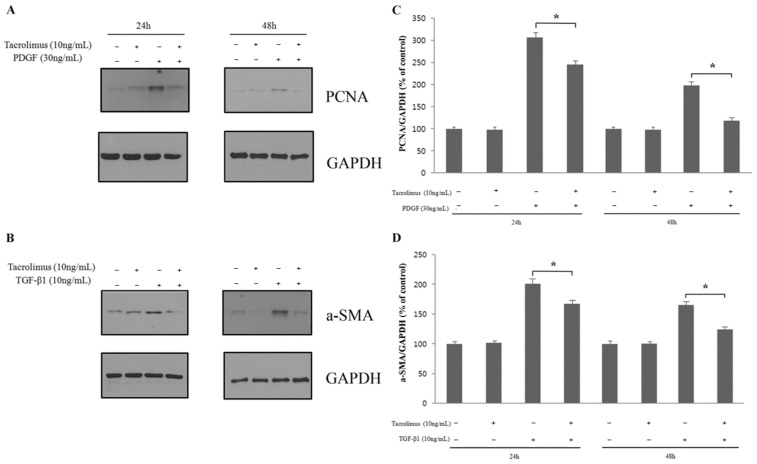
Western blot analysis to identify the proliferation (**A**,**C**) and transdifferentiation (**B**,**D**) of HTFs. The HTF proliferation induced by PDGF was inhibited by tacrolimus (*p* < 0.05; n = 4). The transdifferentiation of HTF cells induced by TGF-β1 was inhibited by tacrolimus (*p* < 0.05; n = 4). *: significant difference between the induced group and tacrolimus-treated induced group ST-1.

**Figure 4 biomedicines-13-02956-f004:**
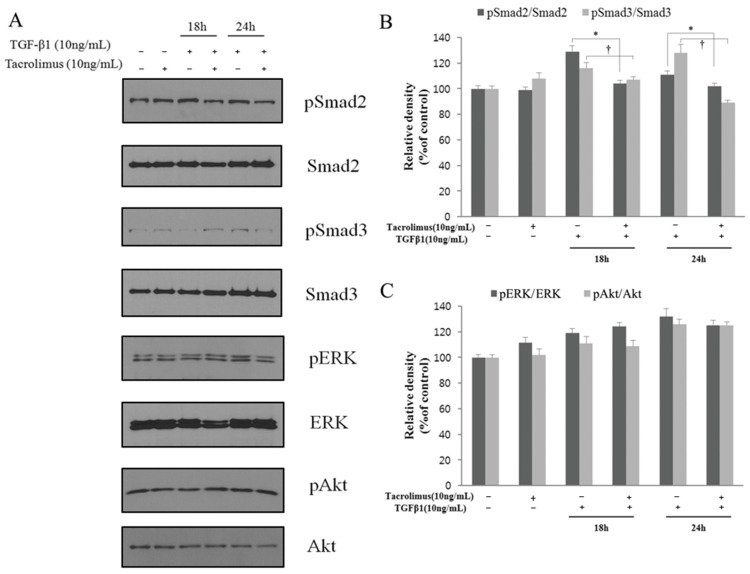
Association analysis of the TGF-β signaling pathway. The levels of pSmad2, pSmad3, pERK, and pAkt were significantly increased in TGF-β1-treated HTF cells (all *p* < 0.05; n = 3) (**A**). Tacrolimus treatment inhibited the pSmad2 and pSmad3 elevation (all *p* < 0.05; n = 3) (**B**), but it did not inhibit the pERK and pAkt increase at 18 h and 24 h (**C**). *: significant difference in pSmad2 between the proliferated group and tacrolimus-treated proliferated group. †: significant difference in pSmad3 between the proliferated group and tacrolimus-treated proliferated group.

**Figure 5 biomedicines-13-02956-f005:**
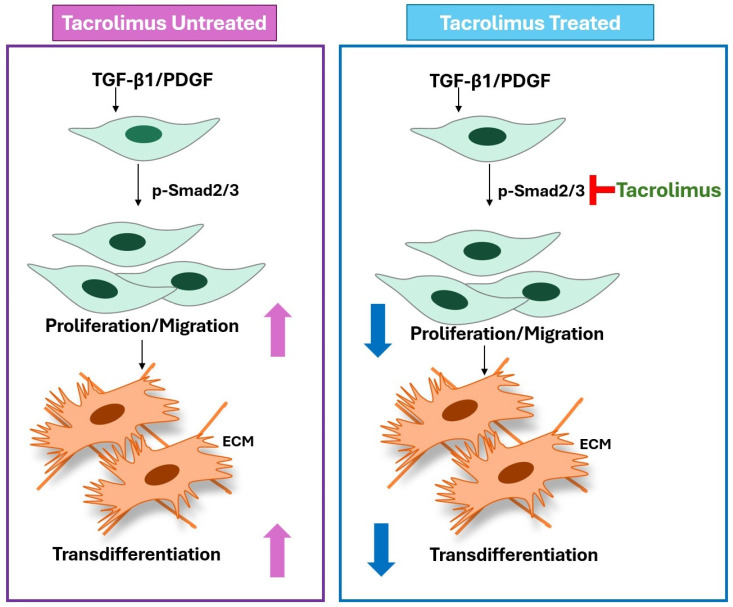
Summary of changes in intracellular signaling and cellular responses in HTF cells after TGF-β1/PDGF stimulation and tacrolimus treatment. TGF-β1/PDGF stimulates p-Smad2/3 activation, thereby promoting proliferation and migration of HTF cells and promoting transdifferentiation (left box, tacrolimus untreated group). Conversely, tacrolimus treatment inhibits p-Smad2/3 activation, thereby inhibiting proliferation, migration, and transdifferentiation (right box, tacrolimus-treated group). The red inhibitor symbol indicates inhibition of the p-Smad2/3 activation pathway by tacrolimus. Upward arrows indicate increased activation or promotion, while downward arrows indicate inhibition or reduction of the process.

## Data Availability

The data presented in this study are available on request from the corresponding author. The data are not publicly available due to privacy and ethical restrictions.

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
