# Peer review of "Tacrolimus Inhibits Human Tenon’s Fibroblast Migration, Proliferation, and Transdifferentiation"

_biomedicines, 2025, doi:10.3390/biomedicines13122956_

Round 1
Reviewer 1 Report
Comments and Suggestions for Authors
This manuscript has valuable data regarding to tacrolimus on the HTF cells. The data are convincing and the interpretation is adequate. However, some concerns should be addressed.
Figure 1. What about tacrolimus only on cell viability? Simply, dose-dependent effect of tacrolimus without PDGF nor TGF can provide valuable information.
Figure 3A. The photo of PCNA at 24 h should be replaced with a better one. First lane and third lane seemed similar band, but C graph showed 3 times of increases. This is not representative.
Reviewer 2 Report
Comments and Suggestions for Authors
Authors presented the manuscript entitled Tacrolimus Inhibits Human Tenon’s Fibroblast Migration, Proliferation, and Transdifferentiation, require major improvement following given below suggestions.
- The introduction should more explicitly state how this study advances the field compared to previous work on tacrolimus and fibroblast inhibition. Emphasize the unique contribution of using human Tenon’s fibroblasts and the potential for clinical translation, especially in comparison to mitomycin C and other antifibrotic agents.
- The discussion section should further elaborate on the limitations, such as the lack of positive controls, the short duration of experiments, and the absence of long-term toxicity data. Suggesting specific future studies would strengthen the manuscript.
- The rationale for selecting specific concentrations of tacrolimus and growth factors should be explained breifly. Additionally, the manuscript could benefit from a brief discussion on the choice of serum starvation and its impact on cell migration and proliferation assays.
- The statistical methods section should specify the exact tests used for each experiment and clarify how multiple comparisons were handled. Mentioning the software version and any corrections for multiple testing would add rigor.
- The discussion could more thoroughly explore the implications of tacrolimus acting primarily through the Smad-dependent pathway, and how this might influence clinical outcomes or therapeutic strategies. Comparing the findings with other calcineurin inhibitors or anti-fibrotic agents would provide context.
- The manuscript should discuss the potential side effects of tacrolimus in the context of ocular use, such as discomfort or systemic absorption, and suggest strategies to mitigate these issues in future clinical applications.
- A visual summary of the proposed mechanism of action of tacrolimus on Tenon’s fibroblasts could help readers understand the key findings and pathways involved.
- The conclusion should emphasize the potential for tacrolimus to be used in slow-release formulations or as a coating for ocular implants, referencing recent advances in drug delivery technologies.
Round 2
Reviewer 1 Report
Comments and Suggestions for Authors
The authors addressed several issues.
Reviewer 2 Report
Comments and Suggestions for Authors
The authors have reflected all the said suggestions and comments, which made the manuscript enhanced with improved readability; Thus, I suggest for further consideration with acceptance.